# The Radiance Differences between Wavelength and Wavenumber Spaces in Convolving Hyperspectral Infrared Sounder Spectrum to Broadband for Intercomparison

**Di Di** [1,2,3] , **Min Min** [3,4,5] , **Jun Li** [6,*] **and Mathew M. Gunshor** [6]

[1] Chinese Academy of Meteorological Sciences, China Meteorological Administration, Beijing 100081, China; didi13@mails.ucas.ac.cn

[2] University of Chinese Academy of Sciences, Beijing 100049, China

[3] Key Laboratory of Radiometric Calibration and Validation for Environmental Satellites (LRCVES/CMA), National Satellite Meteorological Center, China Meteorological Administration (NSMC/CMA), Beijing 100081, China; minmin@cma.gov.cn

[4] School of Atmospheric Sciences and Guangdong Province Key laboratory for Climate Change and Natural Disaster Studies, Sun Yat-sen University, Zhuhai 519082, China

[5] Southern Laboratory of Ocean Science and Engineering (Guangdong, Zhuhai), Zhuhai 519082, China

[6] Cooperative Institute for Meteorological Satellite Studies, University of Wisconsin–Madison, Madison, WI 53706, USA; matg@ssec.wisc.edu

[*] Correspondence: jun.li@ssec.wisc.edu

**Abstract:** Converting the hyperspectral infrared (IR) sounder radiance spectrum to broadband is a common approach for intercomparison/calibration. Usually the convolution is performed in wavenumber space. However, numerical experiments presented here indicate that there are brightness temperature (BT) differences between wavelength and wavenumber spaces in convolving hyperspectral IR sounder spectrum to broadband. The magnitudes of differences are related to the spectral region and the width of the spectral response functions (SRFs). In addition, the central wavelength and central wavenumber should be determined separately in wavelength and wavenumber spaces, respectively; they cannot be converted to each other directly for broadband BT calculations. There exist BT differences (BTDs) between interpolating the resolution of SRF to hyperspectral IR sounder spectrum, and vice versa, for convolution. This study provides clarity on convolution, central wavelength/wavenumber determination, and spectral resolution matching between broadband SRFs and hyperspectral IR sounder radiances for intercomparison/calibration.

**Keywords:** hyperspectral infrared sounder; spectral convolution; intercomparison

## 1. Introduction

Spectral radiance convolution calculation is broadly used in various applications of satellite remote sensing including radiance simulation [1–5], validation [6,7], and space-based intercomparison or intercalibration [8–18]. Generally, the monochromatic radiance spectrum of high spectral resolution calculated from an accurate line-by-line model is regarded as one source of references for calibration and validation of radiometric accuracies of broadband imagers. Mathematically, to simulate radiances observed by a space-based broadband imager, the monochromatic radiance spectrum should be convolved with its SRFs. The objective of spectral convolution is to integrate monochromatic radiance spectrum to match the broadband SRFs and make it comparable with the actual broadband imager radiance observations. Well-calibrated hyperspectral infrared (IR) sounder radiances [19] can also

be used to assess the broadband IR sensors by convolving hyperspectral infrared sounder spectra with their SRFs. Others, such as Gunshor [20] and Wang [15], have used the hyperspectral IR radiance measurements from the (AIRS) and Infrared Atmospheric Sounder Interferometer (IASI) to intercalibrate water vapor bands on broadband instruments such as the Geostationary Operational Environmental Satellite-11 (GOES-11) and GOES-12 imagers. Gong [8] also intercalibrated thermal emissive bands of the Visible Infrared Imaging Radiometer Suite (VIIRS) using Cross-track Infrared Sounder (CrIS) onboard the same platform—Suomi National Polar-orbiting Partnership (SNPP) as a reference.

The spectral radiance convolution formula is mostly given in wavenumber space (see Equation (1)), which has been well documented in various technique reports and articles [6,14,15,21]. In addition, the spectral radiance convolution procedures of the Line-By-Line Radiative Transfer Model (LBLRTM) model are also conducted in wavenumber space. Given a broad IR band, the convoluted band radiance, $\overset{\wedge}{R}$, based on its SRF in wavenumber space, is written as

$$\overset{\wedge}{R} = \frac{\int_{v1}^{v2} rad(v)f(v)dv}{\int_{v1}^{v2} f(v)dv} \tag{1}$$

where $v$ is the wavenumber, $f(v)$ is the SRF in wavenumber space, and $rad(v)$ is the monochromatic or hyperspectral radiance spectrum, which is a function of wavenumber. Considering the fact that SRF of a broadband IR sensor band is often originally measured as a function of wavelength (Xiangqian Wu, personal communication), the spectral convolution can also be calculated in a wavelength space. The Equation (2) below has been taken as a band radiance spectral convolution formula in the wavelength space,

$$\overset{\wedge}{R} = \frac{\int_{\lambda1}^{\lambda2} RAD(\lambda)F(\lambda)d\lambda}{\int_{\lambda1}^{\lambda2} F(\lambda)d\lambda} \tag{2}$$

where $\lambda$ is the wavelength; $RAD(\lambda)$ is the radiance spectrum in wavelength space, which can be transformed from $rad(v)$; and $F(\lambda)$ is the SRF in wavelength space. However, the above-mentioned two equations may bring certain spectral radiance convolution differences in both radiance and brightness temperature (BT), and thus provide influence on satellite radiance intercomparison, intercalibration and validation. The spectral convolution difference (SCD) is investigated in this study.

Determination of central wavelength or central wavenumber of a broad IR band is an important step for BT conversions. The central wavelength and central wavenumber should be determined separately using the same formula in wavelength and wavenumber spaces, respectively, and there are BT differences (BTDs) between these two approaches (in wavenumber and wavelength spaces) for broad IR bands: the magnitude of the BTDs are related to the spectral region and band width of the broad bands.

Another concern is the procedure of spectral resolution matching between broad band SRFs and a hyperspectral sounder when the spectral resolutions are inconsistent. Actually, the spectral resolution of measured SRFs is typically different from that of a hyperspectral IR sounder. There are two typical approaches: (1) the broad band SRF is interpolated to the hyperspectral radiance measurement resolution or (2) the hyperspectral radiance measurement is interpolated to the measurement resolution of the broad band SRF. The question becomes what is the BT difference due to spectral resolution matching or interpolation?

Therefore, the primary objective of this study is to provide understanding/clarity on band radiance spectral convolution, central wavelength/wavenumber determination, along with resolution matching between broadband SRFs and a hyperspectral IR sounder for intercomparison and intercalibration. The equivalent radiance convolution formula is provided and sensitivity experiment schemes for analyzing the spectral radiance convolution difference are explained in Section 2. The BTDs due to different schemes on central wavenumber determination are described in Section 3. The BTDs due to resolution matching between SRFs and a hyperspectral IR sounder are discussed in Section 4. The results of experiments on spectral convolution differences are discussed in Section 5. Summary and conclusions are given in Section 6.

## 2. The Spectral Convolution Differences between Wavenumber and Wavelength Spaces

Symbol descriptions are stated here before formula derivation: $RAD(\lambda)$ and $F(\lambda)$ are the transfer functions of $rad(v)$ and $f(v)$ in wavelength spaces, respectively. Applying the substitution rule for definite integrals to Equation (1) with a substitution function $v = 1/\lambda$, the Equation (1) can be rewritten as

$$\frac{\int_{v1}^{v2} rad(v)f(v)dv}{\int_{v1}^{v2} f(v)dv} = \frac{\int_{\lambda1}^{\lambda2} rad(\frac{1}{\lambda})f(\frac{1}{\lambda})d(\frac{1}{\lambda})}{\int_{\lambda1}^{\lambda2} f(\frac{1}{\lambda})d(\frac{1}{\lambda})} = \frac{\int_{\lambda1}^{\lambda2} RAD(\lambda)F(\lambda)(-\frac{1}{\lambda^2})d\lambda}{\int_{\lambda1}^{\lambda2} F(\lambda)(-\frac{1}{\lambda^2})d\lambda} \tag{3}$$

in which $v_1 = 1/\lambda_1$ and $v_2 = 1/\lambda_2$. According to Equation (3), if using Equation (1) as a standard for convolving, then the equivalent final band radiance spectral convolution formula in wavelength space should be written as

$$\hat{R} = \frac{\int_{\lambda1}^{\lambda2} RAD(\lambda)F(\lambda)(-\frac{1}{\lambda^2})d\lambda}{\int_{\lambda1}^{\lambda2} F(\lambda)(-\frac{1}{\lambda^2})d\lambda} \tag{4}$$

Equation (4) can be similarly rewritten if using Equation (2) as a standard for convolution. It is apparent that Equation (3) shows the inconsistent band radiance spectral convolution formulas in wavelength and wavenumber spaces due to the nonlinear transformation relationship between wavelength and wavenumber spaces. The substitution function between wavelength and wavenumber in the formula uses the international standard unit, and the corresponding adjustment should be made to the substitution function if other units are used.

In comparison between Equation (1)/Equation (4) and Equation (2), it is obvious that there are radiance and BTDs between wavelength and wavenumber spaces (hereafter referred to as spectral convolution difference, or SCD for simplicity). The SCD of a broad IR band depends mainly on the spectral region and its SRF width or band width, as well as radiance intensity. The effects of radiance intensity and SRF width on SCD in wavelength space are relatively clear: the SCD will increase with increasing radiance or with the widening of the SRF band width. However, the quantitative impact of band spectral region and band width on SCD is not readily apparent just from comparison of the two formulas.

To quantitatively analyze the impact of spectral region and SRF width on the SCD between wavelength and wavenumber spaces, several numerical sensitivity experiments were designed and carried out. The Advanced Baseline Imager (ABI) onboard the U.S. GOES-16 satellite [22], and the Advanced Geosynchronous Radiation Imager (AGRI) onboard the Chinese secondary geostationary satellite FengYun-4A (FY-4) [23] are the representatives of broadband radiometers in this study.

The sensitivity experiments are conducted on ten IR bands of ABI, ranging from bands 7 to 16 (from 3.8 to 13.4 µm, shown in Figure 1a), as well as seven IR bands of AGRI from bands 8 to 14 (also from 3.8 to 13.4 µm, shown in Figure 1b), which will be used to demonstrate the dependence of SCD on band spectral region. For analyzing the influence of SRF bandwidth on SCD, using AGRI as an example, the SRFs are simulated as Gaussian functions with a full width half-maximum (FWHM) of 0.25, 0.5, 0.75, and 1 µm, respectively. The simulated FWHM is based on actual SRFs of AGRI IR

bands, of which the largest one is ~0.92 μm. The impact of spectral resolution of hyperspectral IR radiance spectrum on SCD is also discussed. The monochromatic radiance spectra with 0.1 cm$^{-1}$ and 0.001 cm$^{-1}$ spectral resolution are calculated, respectively, using the LBLRTM model with the six standard atmospheric profiles. Considering the effect of atmospheric profiles on SCDs, using the mean values of the absolute radiance and BT differences from the six atmospheric profiles are more meaningful. The radiance spectra from a current hyperspectral IR sounder are also included for understanding the SCD, which is needed for in intercomparison and intercalibration studies. In the sounder/imager comparison experiments on SCDs, IASI radiance spectra simulated from Radiative Transfer for TOVS (RTTOV) model with above-mentioned six atmospheric profiles are also used to quantify the SCDs for AGRI and ABI IR bands. The surface temperature is given as the temperature at the lowest level of the profile level and surface emissivity is set as 0.98 for each standard profile.

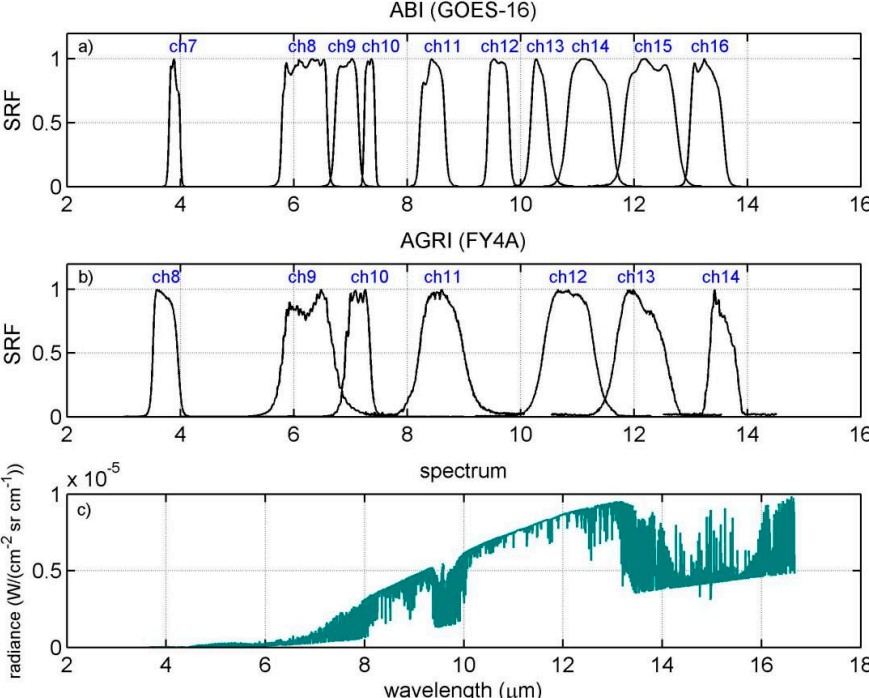

**Figure 1.** Spectral response functions (SRFs) of infrared (IR) bands of (**a**) Advanced Baseline Imager (ABI) and (**b**) Advanced Geosynchronous Radiation Imager (AGRI); (**c**) radiance spectrum of 0.1 cm$^{-1}$ spectral resolution calculated based on the Line-By-Line Radiative Transfer Model (LBLRTM) and the atmospheric profile of mid-latitude summer.

To facilitate an easy interpretation, radiance is commonly converted into equivalent black-body temperatures or alternatively known as equivalent BTs. The simplest approach for converting radiance to BT is to use the Planck function and band central wavenumber [24]. However, this approach is not accurate and additional modifications are required; for example, using band correction at the band central wavenumber with empirical fits of BT against radiance [25]. The band corrections are calculated during prelaunch test by empirical fits of variational black-body temperature and corresponding channel effective brightness temperature at the central wavenumber. For each IR band, the BT (unit: K) can be calculated as

$$T_b = \left\{ \frac{c_2 * v_c}{\ln\left[1 + (c_1 * v_c{}^3)/\hat{R}\right]} - bc_1 \right\} / bc_2 \tag{5}$$

where $\overset{\wedge}{R}$ is the band radiance (unit: mW/m$^2$/sr/cm$^{-1}$); $c_1$ and $c_2$ are the function constants; $c_1 = 1.19104 \times 10^{-5}$ (unit: mW/m$^2$/sr/cm$^{-4}$); $c_2 = 1.43877$ (unit: K/cm$^{-1}$); $v_c$ is the central wavenumber of IR band; and $bc_1$ and $bc_2$ are the fitting coefficients (band correction coefficients). The band correction coefficients used here come from the RTTOV coefficient file for AGRI and ABI imagers. This practical approach is accurate and has been widely used by research community, for example, the GOES-R science team has used this approach for converting spectral radiance to infrared band brightness temperature (https://www.star.nesdis.noaa.gov/goesr/docs/ATBD/Imagery.pdf).

## 3. Determination of Central Wavelength and Central Wavenumber for a Broad IR Band

In the procedure of converting radiance to BT, the band central wavenumber is a prerequisite. When a SRF is given, there are two common ways to determine the band central wavenumber, for example, (1) by convolving with SRF directly in wavenumber domain (Equation (6)) and (2) by transforming from central wavelength, which is convolved in wavelength domain (Equation (7));

$$v_{c1} = \frac{\int_{v1}^{v2} v \cdot f(v) dv}{\int_{v1}^{v2} f(v) dv} \tag{6}$$

$$\lambda_c = \frac{\int_{\lambda1}^{\lambda2} \lambda \cdot F(\lambda) d\lambda}{\int_{\lambda1}^{\lambda2} F(\lambda) d\lambda} \qquad v_{c2} = \frac{1}{\lambda_c} \tag{7}$$

where $v$ is the wavenumber and $f(v)$ is the corresponding measured SRF in wavenumber space; $\lambda$ is the wavelength and $F(\lambda)$ is the SRF in wavelength space. Here, wavelength and wavenumber are both taken as the international standard unit, and conversion coefficient should be adjusted from 1 to 10,000 if common units are taken.

There is a BT calculation difference for a broad IR band in central wavenumber between these two formulas. In order to understand the BTDs in conversion between radiance and BT using the two approaches, a set of experiments was conducted for both ABI and AGRI IR bands. Two sets of band central wavenumbers are determined by Equation (6) and Equation (7), respectively. The radiances of IR bands are simulated from the six standard atmospheric profiles using the RTTOV model. The BTDs for each IR band can be obtained from Equation (5) with two sets of central wavenumbers. Mean central wavenumber and BTDs of six standard atmospheric profiles for the ABI and AGRI IR bands are exhibited in Figure 2. Different wavenumber calculation methods will make notable wavenumber difference and corresponding BTDs in conversion from radiance to BT, especially for water vapor (WV)-sensitive bands. The BTDs are more remarkable for AGRI IR bands than ABI due to relatively broader SRFs of AGRI. The larger BTDs for WV bands may be due to the fact that WV bands are located at shorter wavelengths and have broader SRFs.

In our opinion, the convolution spaces selected for calculating the central wavenumber and band radiance should be the same. In the following experiments, all the central wavenumbers in conversion from radiance to BT are calculated by Equation (6) and the spectral convolution radiances are calculated in wavenumber space.

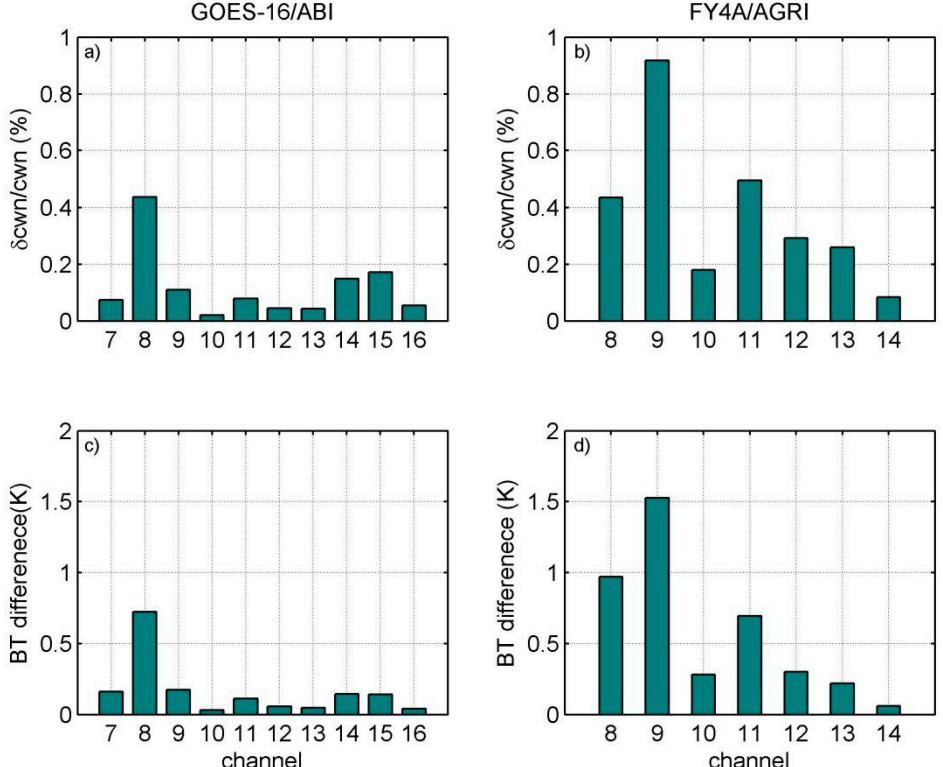

**Figure 2.** Mean (**a**,**c**) central wavenumber differences and (**b**,**d**) central wavenumber caused BT differences (BTDs) from the six standard atmospheric profiles for ABI and AGRI IR bands, respectively.

## 4. Spectral Resolution Matching between SRFs and a Hyperspectral IR Sounder

Equation (8) is the discretization form of the convolution Equation (1), and it implies that spectral resolution of SRFs and a hyperspectral IR sounder should match before the calculation of radiance spectral convolution.

$$R = \frac{\sum\limits_{i=1}^{N} rad(v_i)f(v_i)}{\sum\limits_{i=1}^{N} f(v_i)} \tag{8}$$

As mentioned in the introduction, the spectral interpolation scheme (including interpolation method and interpolation benchmark) remains controversial when the spectral resolution of the measured SRF is inconsistent with that of a hyperspectral IR sounder. Two experiments have been conducted here for selecting an appropriate interpolation scheme for the SCD study. The BTD between interpolating a SRF to a hyperspectral IR sounder, and vice versa, for convolution, is called spectral matching difference (SMD) in this study. The SMD of a broad IR band in BT is calculated in wavenumber space.

The interpolation methods which may be suitable for spectral interpolation are compared in the first experiment. First, interpolation errors are introduced here which are defined as the root mean square differences (RMSDs) between the interpolated function (referring to interpolated SRF or spectrum) and the original function (referring to original SRF or spectrum). It should be pointed out that the same interpolation method need be applied twice if the interpolated function could be comparable with original function. For example, if the SRF is interpolated to IASI's hyperspectral resolution, the new SRF should be applied with the same interpolation method to make it back to the original resolution. In this set of experiments, three commonly used interpolation methods (linear, cubic polynomial, and cubic spline interpolation) are used for SRFs of seven AGRI infrared channels as well as three types of hyperspectral IR radiance spectra (apodized IASI spectra of 0.25 cm$^{-1}$ and

monochrome spectra of 0.1 cm$^{-1}$ and 0.001 cm$^{-1}$). Band mean interpolation errors are then computed to evaluate the merit of interpolation methods. The results for seven AGRI infrared channels are plotted in Figure 3. Due to the IASI spectra not fully covering the first two infrared channels of AGRI, there are no results of channels 8 and 9 when matching the corresponding SRFs with IASI spectra. It is shown in Figure 3 that the cubic spline interpolation method performs best for the SRF interpolation and cubic polynomial interpolation is the best for radiance spectrum interpolation.

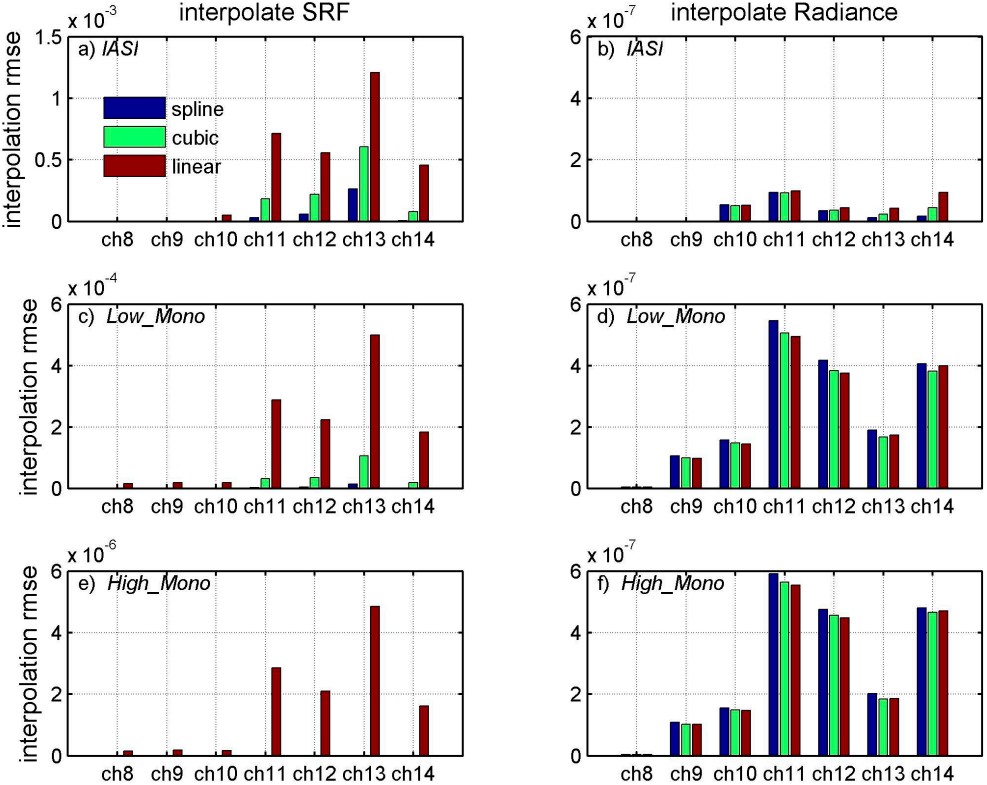

**Figure 3.** Mean root mean square differences from different interpolation methods for 7 AGRI IR bands, hyperspectral sounder radiances with 0.25 (Infrared Atmospheric Sounder Interferometer (IASI), apodized, upper), 0.1 (unapodized, middle) and 0.001 (unapodized, lower) cm$^{-1}$ resolutions, respectively, are used. Six typical atmospheric profiles are used in radiance calculations.

Second, experiments have been conducted here to further compare the performance of two interpolation benchmarks: Scheme 1: Interpolating SRFs to sounder; Scheme 2: Interpolating sounder to SRFs. In these experiments, spectra of various spectral resolutions calculated with six standard atmospheric profiles and LBLRTM models are added. There are six types of hyperspectral IR radiance spectra in total (apodized IASI spectra of 0.25 cm$^{-1}$, monochrome spectra of 0.25 cm$^{-1}$, 0.1 cm$^{-1}$, 0.025cm$^{-1}$, 0.0025 cm$^{-1}$, and 0.001 cm$^{-1}$) for analysis. Note that IASI of 0.25 cm$^{-1}$ is different from monochrome of 0.25 cm$^{-1}$, since the IASI spectrum simulated from RTTOV model has been applied with a Guassian apodization window [6] and its spectrum has been smoothed. First, the mean absolute BTDs between two schemes are calculated from the six standard atmospheric profiles with respect to six types of hyperspectral IR radiance spectra, for AGRI IR bands. The results are shown in Figure 4a. Due to the fact that an IASI spectrum does not fully cover the first two IR bands of AGRI, there are no results for bands 8 and 9 when matching the corresponding SRFs with IASI spectrum. It is apparent in Figure 4a that the SMDs are less remarkable and negligible when the apodized IASI radiance spectra of 0.25 cm$^{-1}$ resolution are used, while the interpolation benchmark will make meaningful or significant SMDs when the higher monochromatic spectral resolution spectrum is used, especially for strong water vapor absorption bands. This result suggests the interpolation benchmark should be stressed even for unapodized hyperspectral IR radiances of lower spectral resolution. However, it also shows

that the broad IR band SMDs are not closely associated with the radiance spectral resolution. Secondly, the standard deviations (STDs) of convoluted BTs for each scheme are calculated, which reflect the dispersion of convoluted BTs. Since the convoluted BTs should align with each other when six types of radiance spectra are matched with the same SRF for a broad IR band, the STD can help to determine the optimal interpolation benchmark. The STDs calculated from the six atmospheric profiles with respect to five types of spectral resolutions for hyperspectral IR sounder are shown in Figure 4b. Only the monochromatic radiances simulated from the LBLRTM are included in the statistics to avoid possible radiance simulation inconsistency between the LRLRTM and RTTOV. It can be seen that scheme 1 (SRF is interpolated to hyperspectral sounder) is more reasonable (with minimum STDs). By the way, the conclusions and findings will not be changed if adding the IASI radiances simulated from the RTTOV into statistics.

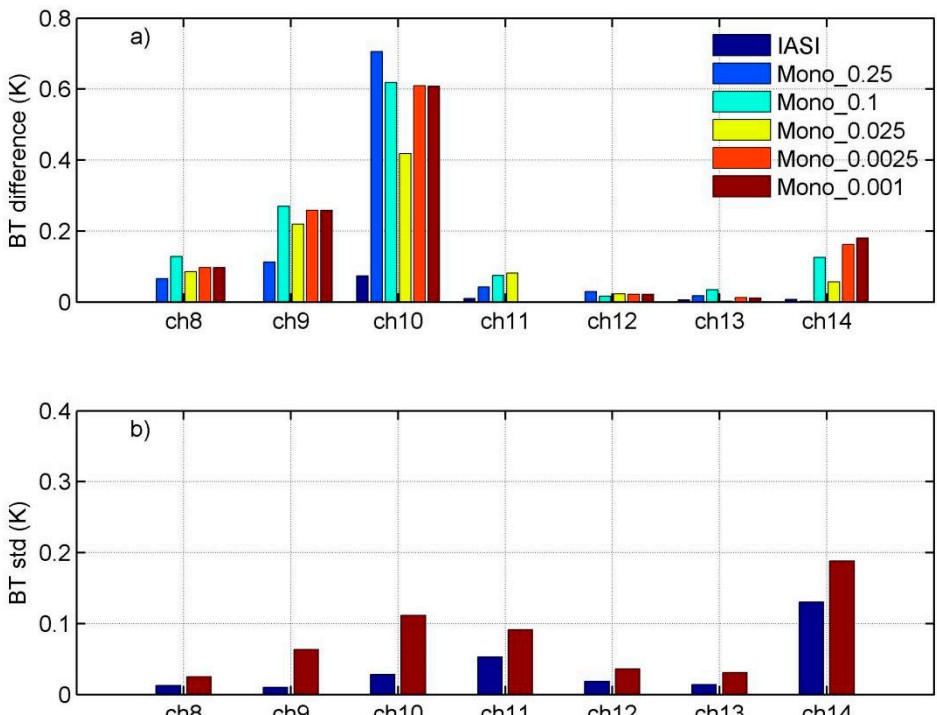

**Figure 4.** (**a**) Mean values of absolute BTDs between the two interpolation schemes calculated from the six standard atmospheric profiles with respect to six different hyperspectral IR radiance spectra (apodized IASI spectra of 0.25 cm$^{-1}$, monochrome spectra of 0.25 cm$^{-1}$, 0.1 cm$^{-1}$, 0.025 cm$^{-1}$, 0.0025 cm$^{-1}$, and 0.001 cm$^{-1}$) for seven AGRI IR bands. (**b**) STDs from convoluted BTs calculated from the six standard atmospheric profiles with respect to three types of IR spectra, using two different interpolation schemes.

In conclusion, the method used to interpolate and match the resolutions will introduce certain error. The interpolation benchmark should be stressed even for unapodized hyperspectral IR radiance spectra of lower spectral resolution. Statistical results show that the approach of interpolating the SRF to the hyperspectral resolution is recommended for most IR bands. Physically, it would always be safe to interpolate the SRF to the spectrum resolution because SRF are generally smoothly varying functions whereas measured IR earth spectra have very sharp changes as a function of wavelength.

## 5. Spectral Convolution Differences

The following experiments in this section have been conducted using the spectral match scheme of interpolating SRF to hyperspectral radiance resolution (scheme 1 from the previous section). SCDs between wavelength and wavenumber spaces are quantified by the radiance and BT differences using wavenumber space as the reference.

Table 1 shows the SCDs calculated by simulated SRFs for seven IR bands of AGRI, in which the SRFs are simulated as Gaussian functions with FWHM of 0.25 μm, 0.5 μm, 0.75 μm, and 1 μm, respectively. It aims to exhibit the influence of broadband SRF width on SCD. It is found that the SCD significantly increases with the SRF width broadening for all bands, while the influence of SRF width on SCD varies in bands. For a constant FWHM of SRFs, the SCDs between IR bands are also noticeable. The effect of hyperspectral bandwidth on SCDs appears to be predictable, with wider bands yielding larger SCDs. The impact of band spectral region on SCD is complicated, and not clearly dependent on longer or shorter wavelength bands. In general, the SCD is larger for the short-wave IR band corresponding to the radiance spectrum where radiance intensity is relatively weak and thus more sensitive to the radiance variance.

**Table 1.** Spectral convolution differences (SCDs) calculated by simulated SRFs for seven IR bands of FY-4A/AGRI with six standard atmospheric profiles, in which the radiance difference is calculated as the percentage (unit: %) of radiance difference to the total radiance as well as BTD (unit: K) using convolution results in wavenumber space as reference.

| Error | Guassian-SRF (μm) | FY-4A/AGRI | | | | | | |
|---|---|---|---|---|---|---|---|---|
| | FWHM | ch8 | ch9 | ch10 | ch11 | ch12 | ch13 | ch14 |
| Δrad/rad (%) | 0.25 | 1.71 | 0.14 | 0.46 | 0.11 | 0.03 | 0.02 | 0.07 |
| | 0.5 | 5.59 | 1.23 | 1.80 | 0.43 | 0.12 | 0.05 | 0.35 |
| | 0.75 | 8.51 | 3.42 | 3.81 | 1.01 | 0.30 | 0.11 | 0.77 |
| | 1 | 10.26 | 6.39 | 6.51 | 1.76 | 0.62 | 0.19 | 1.16 |
| BTD (K) | 0.25 | 0.35 | 0.04 | 0.14 | 0.05 | 0.02 | 0.01 | 0.04 |
| | 0.5 | 1.14 | 0.30 | 0.55 | 0.20 | 0.07 | 0.03 | 0.22 |
| | 0.75 | 1.68 | 0.83 | 1.16 | 0.47 | 0.17 | 0.07 | 0.48 |
| | 1 | 1.99 | 1.57 | 1.98 | 0.81 | 0.36 | 0.12 | 0.73 |

The actual SRFs of AGRI and ABI IR bands (shown in Figure 1) are also included to analyze the realistic SCDs. Mean values of absolute SCDs calculated from the six atmospheric profiles with respect to three types of hyperspectral IR radiance spectra are displayed in Figure 5a (for ABI) and Figure 5b (for AGRI). Both of the results show almost no influence of spectral resolution of radiance spectrum on SCDs. Similarly, the SCDs are relatively larger for the short-wave IR and some water vapor bands. Due to the fact that the FWHMs of ABI IR band SRFs are comparatively smaller than those of AGRI, the SCDs for ABI are also smaller. For FY-4A/AGRI, almost every IR band has relatively large (greater than 0.1 K) SCD. However, for ABI IR bands with narrow bandwidth, SCDs are small (less than 0.1 K) for all but two short-wave IR bands and a long-wave IR band. The SCDs are significant for various broadband imagers, and they should be taken into account for intercomparison, intercalibration, and validation using hyperspectral IR sounder spectra. The prelaunch goal for ABI IR band noise (at 300 K) is 0.1 K for bands 7–15 and 0.3 K for band 16, and on-orbit performance is even better than that. It would be preferable for the convolution process errors to be less than the instrument noise.

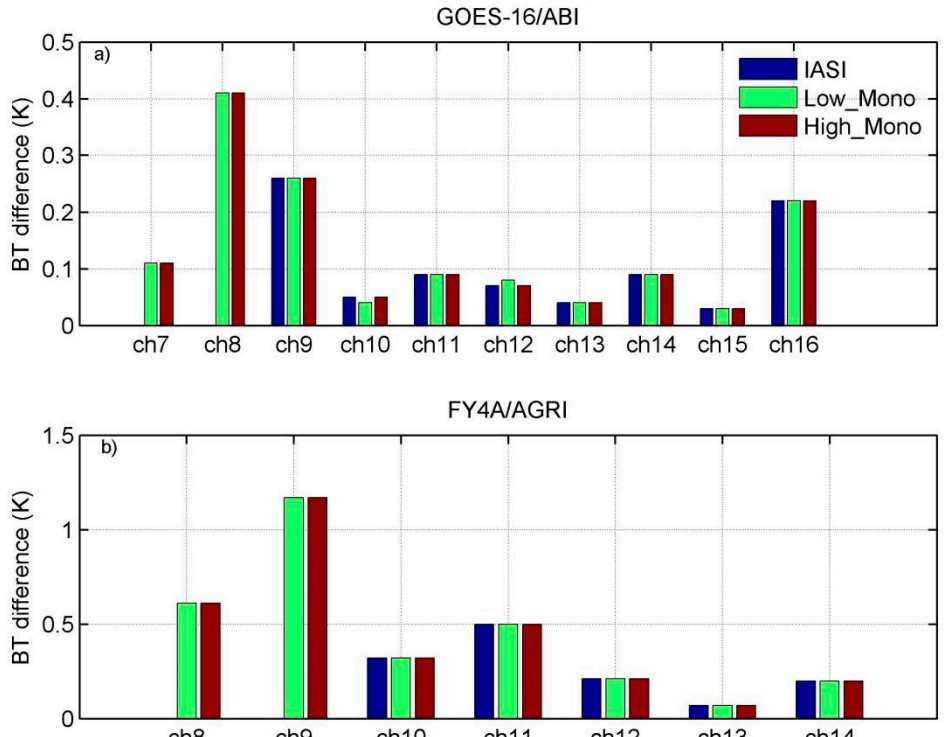

**Figure 5.** Mean value of SCDs calculated from the six standard atmospheric profiles with respect to the radiance spectrum with three different resolutions ($0.25\ cm^{-1}$, $0.1\ cm^{-1}$, and $0.001\ cm^{-1}$) for (**a**) ABI and (**b**) AGRI IR bands.

## 6. Summary and Conclusions

There are brightness temperature differences (BTDs) between wavelength and wavenumber spaces in convolving hyperspectral IR sounder spectra to broad IR bands, those BTDs should be taken into account for intercomparison, intercalibration, and validation. Through numerical analysis, the following was found.

(1) The spectral convolution difference (SCD) has a direct relationship with spectral response function (SRF) width of broadband, and for a given spectral region, with the wider SRF yielding a larger SCD. The impact of SRF band width on SCD varies by band (spectral coverages).

(2) The spectral resolution of the hyperspectral radiance spectrum has little influence on SCD.

(3) The SCDs are relatively larger for the short-wave IR band than other IR bands. It may be attributable to the corresponding radiance values in the short-wave IR region being relatively smaller than in the longer-wave IR bands, and thus more sensitive to the radiance variance.

(4) The spectral resolution match procedure involved in spectral convolution when the spectral resolution of SRF is inconsistent with hyperspectral radiance is also discussed. Interpolation method and benchmark should be selected prudently even for unapodized hyperspectral IR radiance of lower spectral resolution. The scheme of interpolating the SRF to the hyperspectral resolution is recommended.

(5) The central wavelength and central wavenumber should be determined separately using the same formula in wavelength and wavenumber spaces, respectively, and there are BTDs between these two approaches for broad IR bands; the differences are related to the spectral region and band width.

**Author Contributions:** Conceptualization, J.L. and M.M.G.; methodology, D.D.; software, D.D.; validation and verification of results, D.D.; formal analysis, D.D.; investigation, D.D. and M.M.; resources, M.M.; data curation, D.D.; writing—original draft preparation, D.D.; writing—review and editing, J.L. and M.M.G.; visualization, D.D.; supervision, J.L.; project administration, M.M.; funding acquisition, J.L., M.M.G. and M.M.

**Funding:** This research was funded by the National Key R&D Program of china 2018YFB0504800 (2018YFB0504802) and NOAA JPSS and GOES-R series science program NA15NES4320001.

**Acknowledgments:** The SRFs of FY-4A/AGRI and GOES-16/ABI infrared channels were obtained from the National Satellite Meteorological Center (NSMC) of China Meteorological Administration (CMA) and NESDIS/NOAA (https://ncc.nesdis.noaa.gov/GOESR/ABI.php), respectively. The authors are particularly grateful to Christopher C. Moeller for his review and helpful comments on this paper.

**Conflicts of Interest:** The authors declare no conflict of interest.

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
