# Peer review of "The Radiance Differences between Wavelength and Wavenumber Spaces in Convolving Hyperspectral Infrared Sounder Spectrum to Broadband for Intercomparison"

_remotesensing, doi:10.3390/rs11101177_

Round 1
Reviewer 1 Report
During inter-calibration/inter-comparison for atmospheric sounding, hyper spectral resolution sounding profiles (such as IASI) are often used as reference to calibrate broadband spectra sounding, such as GOES ABI sounding channels. This paper discussed the impact of different numerical schemes (mainly different interpolation resolution) on the convolution results from hyperspectral resolution radiance to broadband radiance and remind readers the difference of the central wavenumber and central wave-length difference.
Clearly, from the high resolution spectra to broad band convolution, the numerical scheme used is critical to obtain a correct result (radiance or brightness temperature). Any scientist/engineer engaging with calibration should be aware of this error when assessing their results. The thumb of rule for the convolution is to interpolate the coarse resolution to higher resolution whenever large variability can be expected either in wave number or wavelength space. To assess of the results, one should consider the difference of the interpolation schemes, such as linear versus cubic with as high resolution as possible, instead of high/low resolution comparison, because low resolution simply brings larger error especially when the actual spectra has high variability.
Therefore this study is not new or original. I do not recommend this paper to be published at Journal of Remote Sensing.
Other comments:
1. ) Equation should start from Plank Function to illustrate why there is an additional term in radiance conversion (square of wavelength terms) from wave number space to wave length space. The equations (e.g. 7) have vague meaning, especially for radiance.
2. ) The central wave number and central wavelength can only be used in their own space to corresponding radiance. Direct conversion from Plank function is only to be used as a coarse evaluation of the results, not for accurate comparison; Corrections must be added for accurate conversion from radiance to brightness temperature using central wavenumber/central wavelength, or using a look up table between radiance and brightness temperature. Those correction coefficients should be different depending on using wavenumber or wavelength space. However, it should be determined during pre-launch test whether wavenumber or wavelength should be used. The author did not mention or do not know how these correction terms can be obtained from Plank Function.
3. The authors only mention high/low resolution comparison but did not mention any interpolation schemes used (cubic/linear etc). However the cubic interpolation/linear interpolation can cause different results. This part is vague in the paper and not clearly shown.
4. It would be better to use a discretization form of the convolution equation instead of an integration This way, it will show where the errors are from clearly and hence the author can have an error budget based on the numerical schemes.
Author Response
We appreciate your time and reviewing our manuscript. We have revised our manuscript accordingly. Please see the attachment below.

Reviewer 2 Report
The manuscript entitled "The Radiance Differences between Wavenumber and Wavelength Spaces in Convolving Hyperspectral Infrared Sounder Spectrum to Broadband for Intercomparison" by Di Di, Min Min, Jun Li and Mathew M. Gunshor is very well structured and very well written. It is well-known that broad band channels bring about difficulties in defining conversions from radiance to brightness temperatures and radiative transfer codes like RTTOV use band correction coefficients for this purpose. Inter-comparing hyperspectral infrared sounders with more traditional broadband infrared satellite spectrometers is of great importance for calibrating and validating new instruments. In these inter-comparisons matters become more difficult as hyperspectral interferometer instruments and their SRFs are characterised in wavenumber space, while some of the more traditional infrated spectrometers would be characterised in wavelength space. The authors demonstrate in a very methological way how comparisons involving wavenumber versus wavelength space have to be done and how the inter-comparison of brightness temperatures from hyperspectral and broadband is effected. They also nicely quantify the differences introduced when making inter-comparisons. The paper addresses issues which are very relevant in practical applications and will benefit these. I sincerely recommend the publication of this paper as it is.
Author Response
We appreciate your time and reviewing our manuscript. Thanks for your inspiring comments.
Reviewer 3 Report
This is an interesting work, which is helpful to under some common errors in convolving hyper-spectral measurements to broad bands. I would like authors to comment in terms of wavelength and wavenumber, which of the two provides more accurate estimate of radiance or brightness temperature?
Author Response
We appreciate your time and reviewing our manuscript. Thanks for your inspiring comments.
Regarding the convolution space, in our opinion, convolving hyper-spectral measurements to broad bands in wavenumber space is more accurate.
Reviewer 4 Report
Generally, the paper is technically correct, but hard to follow. The methods part and the results part are mixed together, which makes the paper like a work report.
Converting the hyperspectral infrared (IR) sounder radiance spectrum to the broad bands using spectral response functions (SRFs) is a common approach for inter-comparison/calibration. This paper illustrates the brightness temperature (BT) differences between wavelength and wavenumber spaces in convolving hyperspectral IR sounder spectrum to broad bands by numerical analysis. However, the research is preliminary and lacks innovation.
(1) In this paper, RTTOV and LBLRTM are used. RTTOV is a rapid radiation transfer model, whose accuracy is not as good as LBLRTM. Will the consistency between the two models affect your later results?
(2) It is well known that interpolating the spectral response function to the resolution of the radiance gives better results than interpolating the hyperspectral radiance to the spectral response function. Please explain the anomaly of band 14 in figure 3.
(3) It is not accurate to calculate the BT using the equivalent central wavelength (wavenumber) using equation 5. Does it make sense to compare the difference of calculated the BTs?
(4) The land surface conditions (land surface temperature and land surface emissivity) are not specified.
(5) The standard atmospheric profiles are not enough, at least an atmospheric profile database like TIGR should be used to quantify the differences.
Therefore I would like to suggest rejecting this paper at this time and resubmitting later.
Author Response

(The authors gave the same response as above.)

Round 2
Reviewer 1 Report
I appreciate the author took efforts to revise the paper. I have one question on the revised paper.
About Eq.1 and Eq 4, if your definition of RAD(lambda) in both Eq (2) and Eq. (4) are consistent, then what's the meaning of Eq. (4)? What's the unit of radiance? If you replace the RAD(lambda) using the plunck function expression of radiance in wavelength space (per wavelength) into Eq. (4), will you expect the radiance in wavelength space or wavenumber space? Can you show some code to see how RAD(lambda) is used/expressed?
Minor comments: 1.) Missing a dv term in equation 6.)
Author Response
Thank you very much for your time and reviewing our manuscript.
Our responses to your comments are attchached.

Reviewer 4 Report
The authors have revised the manuscript to some extent,but I do not think the authors have addressed my concerns. They are tried to explain that what they did is reasonable.
Author Response
Thank you very much for your time and reviewing our manuscript. Your comments are helpful and improve our works. We have done our best to address all your concerns/comments. Please allow us explain our modifications based on your comments.
(1) Your first two comments are addressed seriously in the revised manuscript.
(2) Your third comment is questioning the conversion formula from radiance to BT. Probably we did not state clearly. If converting radiance to BT with central number and without addition modifications, the results are not accuracy. However, the addition modifications have actually already been used in this study, it has enough computational accuracy and also it is the common method used by research community.
(3) The land surface conditions are specific in the revised manuscript as suggested. Our research focuses more on analyzing and clarifying potential issues/problem on convolving hyperspectral IR radiances to broadband by applying SRFs, especially when different approaches are employed in the convolution.
(4) Your last comment is also addressed seriously in the revised manuscript. The results from 83 atmospheric profiles (close to reality) are reproduced. To us, the new results do not change our conclusions and findings. Therefore, we remain the six typical atmospheres in our calculations.
Besides revisions based on all Reviewers’ comments including yours, we have also made significant changes listed below:
(5) We have added more analysis on different methods for interpolation between SRFs and a hyperspectral IR sounder, which is useful for inter-comparisons between broad IR bands and hyperspectral IR sounder.
(6) We have two native English speakers to revise/improve English, so the revised manuscript should be easier to be understood now.
We hope the above changes along with our previous response/revision help.